# The interplay between cognitive biases, attention control, and social anxiety symptoms: A network and cluster approach

**Nathalie Claus**◯*, **Keisuke Takano**◯, **Charlotte E. Wittekind**

Department of Psychology, Division of Clinical Psychology and Psychological Treatment, LMU Munich, Munich, Germany

* nathalie.claus@psy.lmu.de

## Abstract

Cognitive models of social anxiety highlight the importance of different cognitive biases (e.g., attention bias, interpretation bias) and executive dysfunctions, which have, however, mostly been investigated in isolation. The present study explored their interplay using two statistical approaches: (1) network analysis to identify the unique associations between cognitive functions, and (2) cluster analysis to reveal how these associations (or combinations) are manifested in a population. Participants from the general population ($N$ = 147) completed measures of attention control, attention bias, interpretation bias, and social anxiety symptoms. Network analysis showed an association between social anxiety symptoms and interpretation bias, although no other significant associations emerged. Cluster analysis identified a group of participants characterized by an adaptive cognitive pattern (i.e., low cognitive biases, good executive function); and a group exhibiting a more maladaptive pattern (i.e., high interpretation bias, good alerting but poor executive function). The maladaptive group showed higher levels of social anxiety than the adaptive group. Results highlight the strong association between social anxiety symptoms and interpretation bias, while challenging the putative role of attention bias. Attention control, particularly executive function, may limit the impact of cognitive bias on anxiety symptoms.

## Introduction

### Cognitive biases and attention control in social anxiety

Cognitive theories of social anxiety highlight the role of cognitive dysfunctions in the etiology and maintenance of the disorder [1]. Social anxiety is characterized by intense fear of one or several social situations in which individuals may be observed and negatively appraised by others [2]. A striking feature of social anxiety is that the fear persists although socially anxious individuals cannot fully avoid social situations and often do not receive any negative feedback in these situations [3, 4]. Cognitive models explain this phenomenon by stressing the relevance of biased or impaired information processing. For example, attention bias toward threat is thought to impede habituation; additionally, impaired attention control may exacerbate this effect [5].

**Data Availability Statement:** Raw data and R code are available at DOI 10.17605/OSF.IO/YUE4S.

**Funding:** Keisuke Takano was supported by Alexander-von-Humboldt foundation. The funders had no role in study design, data collection and

analysis, decision to publish, or preparation of the
manuscript.

**Competing interests:** The authors have declared
that no competing interests exist.

In the context of anxiety, two types of cognitive biases are particularly relevant: biases in attention and interpretation. *Attention* bias in social anxiety is typically operationalized as faster engagement to [6] or slower disengagement [7, 8] from socially threatening stimuli relative to neutral ones. Threatening stimuli may be angry or disgusted faces or words related to embarrassment and shame [4, 9]. In the context of social anxiety, facial expressions are considered to be relevant and ecologically valid stimuli [10, 11] as they contain information about potential judgement by other people [12]. Different paradigms have been used to explore attention biases, such as the Emotional Stroop Task [13], the Emotional Spatial Cueing Task [14, 15], the Dot Probe Task [16], and the Visual Search Task [17, 18]. Although there is an ongoing debate on the validity and reliability of these paradigms [19], the Visual Search Task (VST) is known to be a more reliable measure of attention bias in social anxiety [9, 20]. It requires participants to quickly detect a threatening face surrounded by neutral faces. Individuals with social anxiety symptoms have been shown to engage more quickly with threatening relative to neutral faces [20–22] and angry compared to happy faces [10, 23], implying an attention bias for social threat. However, recent studies suggested that this effect may be smaller and less robust than previously assumed [6, 24, 25].

*Interpretation* bias is particularly relevant to social anxiety as information in social contexts is often ambiguous [26]. Neutral or even positive comments or facial expressions can easily be interpreted as negative or threatening, e.g., a smile as an indicator that one is being made fun of [26]. Such negative interpretations have been consistently found in social anxiety [27] both at clinical and subclinical levels [28]. Interpretation biases have been assessed using different paradigms, such as the Ambiguous Scenario Task [29–32], the Sentence Completion Task [33], and the Scrambled Sentence Task [34–36]. The Scrambled Sentence Task (SST) asks participants to build complete emotional sentences, by using five out of six words which can result in either a positive or negative sentence. The SST is thought to be less influenced by conscious control (e.g., social desirability) than other interpretation-bias tasks.

The biased cognition found in social anxiety can be linked to individual differences in *attention control* [37]. Research has shown that attention control acts as a possible moderator between cognitive biases and psychopathology [38–40]. Deficient attention control has been observed in individuals with social anxiety [5] in relation to emotional [41] as well as non-emotional stimuli [42, 43]. Researchers have suggested a model of attention [44, 45] with three components of attention control: (1) the alerting network, which facilitates sensitivity for new stimuli and preparedness to react; (2) the orienting network, which selects information by engaging or disengaging attention; and (3) the executive function network, which controls attention and solves conflicts between reaction alternatives. In anxiety (not limited to social anxiety), evidence is relatively consistent that anxiety is positively associated with the alerting and negatively associated with the executive function network [46–48]. However, the role of attention control in social anxiety is less clear. Whereas symptoms of social anxiety correlate positively with the alerting network [49], they are either negatively [42, 43] or positively associated [49] with the orienting network. Furthermore, social anxiety symptoms were found to have a negative [46] or null association [49] with the executive function network. These mixed findings may be explained by the fact that many studies measured only single components of attention control [42].

Although a number of studies have provided empirical (even if inconsistent) evidence for the associations between social anxiety and different cognitive biases, these biases have typically been investigated in isolation [37, 50, 51]. Therefore, it is largely unknown if (and how) different types of cognitive biases and dysfunctions are related to each other. Researchers have called for more comprehensive investigations on multiple types of cognitive (dys)functions and combined cognitive biases in social anxiety [26]. This approach would clarify whether

various cognitive biases interact with and adversely influence each other. There is preliminary evidence for such an interaction in depression [52]. For instance, it has been shown that biased interpretation mediates the association between biases in attention and memory [53] and that a combination of several biases exacerbates symptoms [54]. As of yet, such empirical evidence with regard to social anxiety is still scarce [55–57]. The current study explored such interplay between cognitive biases and attention control with a particular focus on attention and interpretation bias as well as three attentional networks (i.e., alerting, orienting, and executive attention). Understanding these interdependent connections between different cognitive (dys) functions is crucial in understanding the maintenance of social anxiety.

## The present study

The aim of the current study was twofold. First, we investigated associations between social anxiety symptoms and different cognitive measures encompassing attention bias, interpretation bias, and components of attention control, using network analysis. As such, this is a conceptual replication of Heeren and McNally [56], who performed network analyses on a set of cognitive measures and social anxiety symptomatology. They found that fear and avoidance of social situations as well as the orienting component of attention control are the most central variables, i.e., the variables that had the strongest associations with other variables in the network. However, the expected associations between anxiety and attention biases did not emerge. There are several possible reasons for these null associations. For example, the researchers used a Spatial Cueing Task as a measure of attention bias, which has been criticized for its poor psychometric properties [58]. Additionally, the task used verbal stimuli, which may be less effective to cause attention capture than pictorial stimuli [11]. To remedy these limitations, the current study used a VST with facial stimuli. Another important addition was the use of the SST to assess interpretation bias, which allowed us to inspect how interpretation bias is associated with attention bias and different components of attention control in the psychological network of social anxiety.

Second, we explored the interaction between multiple cognitive features, for which we used hierarchical cluster analysis on the multiple cognitive measures. This analysis informs how individuals can be grouped on the basis of their profiles of cognitive measures, whereas the network approach visualizes the proximity (i.e., correlations) between cognitive measures in a given population. The unique advantage of the cluster analysis is that this analysis clarifies if there are groups of individuals who possess single vs. multiple cognitive dysfunctions and if this group difference is related to social anxiety.

Although these two types of analyses are exploratory by nature, we hypothesized that the network analysis would yield positive associations between social anxiety symptoms and attention bias, between social anxiety symptoms and interpretation bias, and between attention bias and interpretation bias. Also, we expected negative associations between social anxiety symptoms and components of attention control, and between attention bias and components of attention control. For the cluster analysis we expect to identify a group of individuals with multiple cognitive biases (e.g., attention and interpretation biases), who would show higher levels of social anxiety symptoms than other groups (with single or no cognitive bias).

## Material and methods

### Participants

We performed a priori power analysis to determine the sample size. We first reviewed published studies to find a good prior for the correlations between the cognitive measures and social anxiety (see S1 Table in S1 File). Second, we simulated data from the identified

correlations with varying sample sizes, and for each simulated dataset, we tested whether each edge was identified through a partial-correlation-network analysis under the assumption of alpha = 0.05 applying Bonferroni correction. With 500 iterations, the power to detect the edge between measures of interpretation bias and social anxiety achieved 0.83 for $N = 150$.

In an attempt to increase variance of observed anxiety symptomatology, participants were recruited from the general population. Inclusion criteria were: age > 18 years; no lifetime diagnosis of a severe neurological disorder (e.g., Multiple Sclerosis, Epilepsy); normal or corrected-to-normal vision; and good knowledge of the German language.

Participants ($N = 157$; 121 women) were recruited through printed advertisements on the university campus and local supermarkets. The study was advertised digitally via a mailing list offered by LMU Munich and recruitment groups on Facebook. The mean age was 23.41 (SD = 7.07) years. Most participants (59%) were university students, with 29% having already obtained a university degree. Undergraduate psychology students received course credit for their participation; alternatively, participants were offered a lottery to win one of five Amazon vouchers (worth €25 each).

Data of seven participants were excluded (six had technical issues; one was not fluent in German), resulting in a sample of $N = 150$. Out of those, three additional participants had to be excluded due to an accuracy rate lower than .80 in the Attentional Network Task.

All participants provided written informed consent prior to participation. The study protocol (57_Takano_b) was approved by the Ethics Committee of the Department of Psychology at LMU Munich.

## Measures

Additional information on the measures used can be found in the S1 File.

### Social anxiety symptoms: Social phobia inventory

To measure symptoms of social anxiety, the German version [59] of the Social Phobia Inventory [60] was administered. The questionnaire consists of 17 self-rated items. It measures severity of central symptoms of social anxiety within the last week on a five-point Likert scale, with possible scores ranging from 0 to 68. For both clinical and subclinical samples, the Social Phobia Inventory (SPIN) has been shown to be internally consistent ($\alpha = .95$) and highly correlated (between $r = .80$ and $r = .88$, $p < .01$) with other social anxiety scales [61]. Cronbach's alpha for the current sample is $\alpha = .72$, with scores ranging from 10 to 48 and 33% of the sample scoring above a cut-off of 25 [61], see S1 and S2 Figs in S1 File.

### Interpretation bias: Scrambled sentence task

The Scrambled Sentence Task [62, 63] measures interpretation biases. Participants are presented with scrambled sentences consisting of six boxes each containing one word (e.g., "nervous don't groups very me make"). Out of these six boxes, five must be selected by clicking on them as quickly as possible (within 10s) to build a grammatically correct sentence, which can either be negative (e.g., "groups make me very nervous") or positive (e.g., "groups *don't* make me nervous"). Participants are instructed to build the sentence that first comes to their mind (Fig 1). As soon as participants have clicked on a word, their response cannot be corrected. In the current study, the task consisted of five practice trials and 20 experimental trials (10 neutral sentences, 10 emotional sentences) presented in random order, with sentences presented at 1.8% of screen in height. At the beginning of the Scrambled Sentence Task (SST), participants were told to respond as quickly and accurately as possible. As in previous SST studies [53], a cognitive load task was added to avoid deliberate response strategies. Before the first

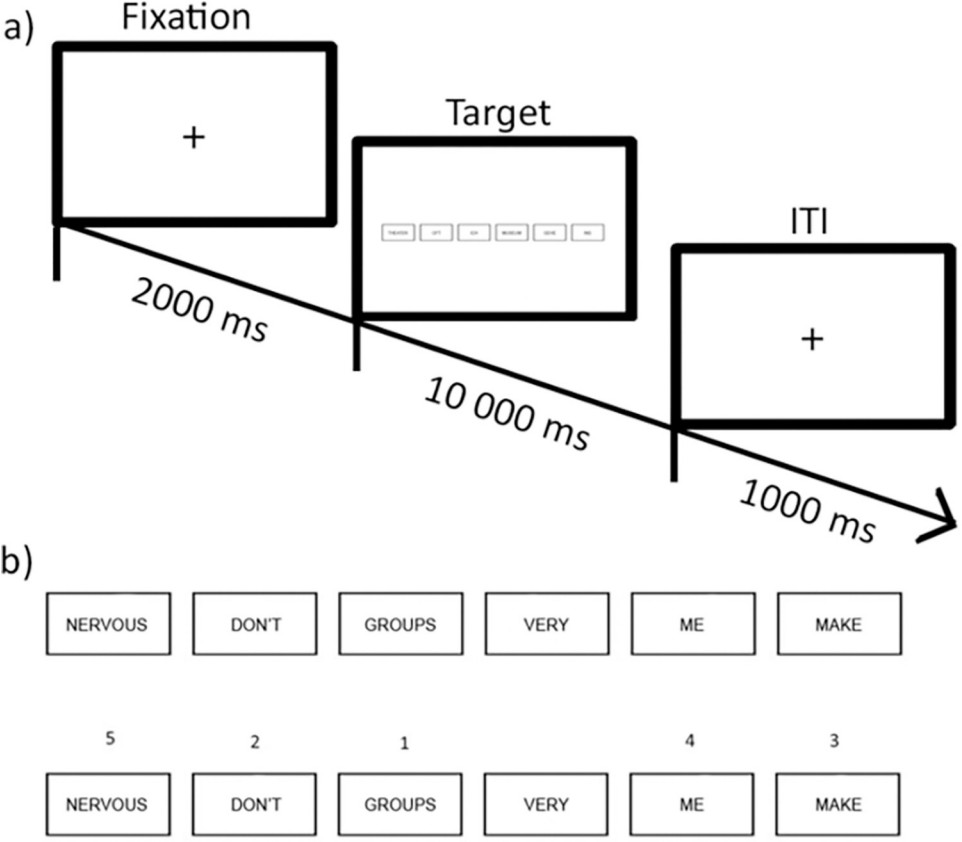

**Fig 1. Experimental set-up of the SST.** Schematic flow of a trial in the Scrambled Sentence Task (a) and example Target displays (b). Participants are presented with a scrambled sentence (see top line of Panel b) and required to click on five of the six words in an order that produces a grammatically correct sentence (see bottom line of Panel b). In this display, the sentence ("nervous don't groups very me make") is unscrambled in a positive fashion ("groups don't make me nervous").

experimental trial, participants were told to memorize a 6-digit-number (presented for 7000ms) which they were asked to recall at the end of the test.

Since the SST was originally developed for interpretation bias in depression, the current study adapted the stimulus set for social anxiety. Based on the DSM-5 criteria for social anxiety [2] as well as the SPIN [60] and the Social Interaction Anxiety Scale [64], new sentences relevant to socially anxious symptomatology were developed. Neutral sentences were taken from the original German stimulus set for depression.

For data analysis, a negative bias in interpretation was calculated as the ratio of negatively completed sentences divided by the total number of correctly completed emotional sentences, with a higher score indicating a greater bias. Split-half reliability with odd-even trials was $r =$ .35 in our data ($N = 150$). Incorrect trials (954 trials, 31.8%) were excluded, as well as latencies below 1000ms (zero trials, 0%) and above 10000ms (52 trials, 1.6%).

### Attention bias: Visual search task

Attention bias was assessed using the VST [65, 66]. In this task, participants are instructed to detect a target among distractors as quickly as possible, i.e., to find one divergent stimulus among an array of identical stimuli. When presented with an array of stimuli arranged around

a fixation cross, participants must perform a detection task to determine whether a display includes a divergent stimulus (target-present trial) or whether all stimuli within the array are identical (target-absent trial). Similar to previous studies using a VST [67–70], the current study used an array of eight faces arranged in a square and required participants to press the "Y"-key (for "yes", on a QWERTZ keyboard) on target-present trials or the "N"-key (for "no") on target-absent trials. Participants were asked to respond as quickly and accurately as possible. Two thirds of all trials were target-present trials, trials were presented in random order. Within each trial, only faces of one of four models were used so that only the facial expression could differ. The VST follows a 2x2x3 factorial design (target-present vs. target-absent; facilitation vs. interference condition; happy vs. angry vs. neutral target).

Target-present trials either consisted of one emotional target (happy or angry) surrounded by neutral distractors (facilitation condition) or one neutral target surrounded by identical emotional distractors (interference condition). Target-absent trials consisted of either happy, angry or neutral faces only. To control for order effects, one half of participants received two blocks of the interference condition first (Version A), while the other half started with two blocks of the facilitation condition (Version B). After 20 practice trials, participants received a total of four blocks of 48 trials each, resulting in a total of 192 trials. They received error feedback throughout the entire task. Fixation and inter-trial intervals were set to a default, while each array was presented until a response was given (Fig 2).

Similar to previous studies [71], two scores were calculated: A disengagement score was calculated by subtracting RTs of interference trials with happy faces from RTs of interference trials with angry faces (RT interference angry–RT interference happy), i.e., a higher score indicated greater difficulty disengaging from angry relative to happy faces. An engagement score was calculated by subtracting RTs of facilitation trials with angry faces from RTs of facilitation trials with happy faces (RT facilitation happy–RT facilitation angry), i.e., a higher score indicated greater vigilance towards angry relative to happy faces. Since these scores use target trials only, non-target trials were removed. Odd-even reliability for the current sample was $r =$ .12 for the engagement and $r =$ .08 for the disengagement score ($N =$ 150). Despite the low reliability we decided to run the analyses as planned. For analysis, incorrect trials (2208 trials,

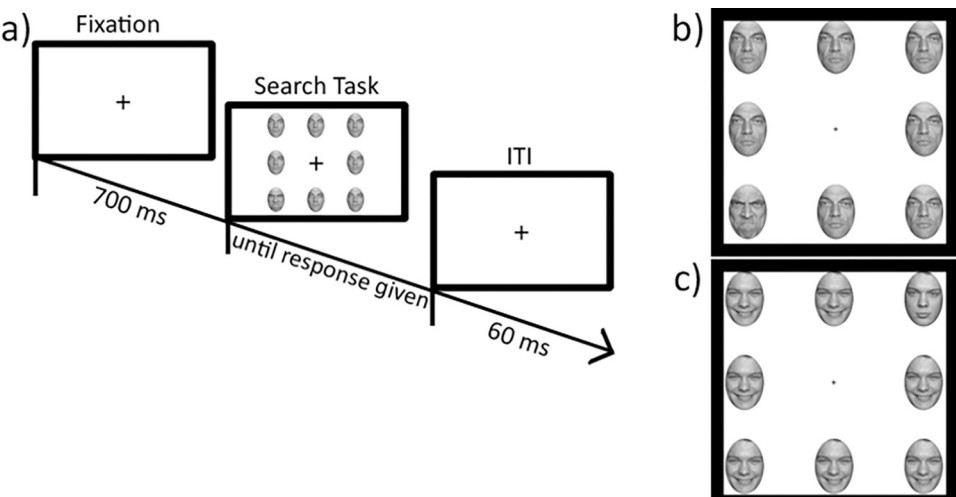

**Fig 2. Experimental set-up of the VST.** Experimental trial (a) and example "search task" arrays of the facilitation (b) and interference (c) conditions of the VST. Participants are presented with an array of eight faces and required to detect whether there is a divergent target present or whether all faces are the same. Both example arrays represent target-present trials.

11.2%) were excluded, as well as latencies below 200ms (2 trials, < 0.01%) and above 2000ms (2137 trials, 10.3%). Additionally, trials 3 SDs above or below each participant's mean were discarded as outliers (21 trials, < 0.01%).

## Attention control: Attentional network task

The Attentional Network Task [72] measures the three components of attention according to Petersen & Posner [44]: i.e., alerting, orienting, and executive function. For a detailed description of the Attentional Network Task (ANT), see S1 File. In this current study, participants went through 24 practice trials, followed by three blocks of 96 trials each, resulting in a total of 288 trials. Before each block, participants were asked to respond as quickly and accurately as possible.

For data analysis, three scores of attention control were calculated: An alerting score was calculated by subtracting RTs of trials with no cues and trials with double cues. An orienting score was calculated by subtracting RTs of trials with center cues and trials with spatial cues. An executive function score was calculated by subtracting RTs of trials with incongruent flankers and trials with congruent flankers. Higher score values each indicate greater attention control capacities. Consistent with other studies [72, 73], only correct responses were included in our statistical analyses. Odd-even reliability for this current sample is at $r = .32$ for the alerting score, $r = .05$ for the orienting score, and $r = .46$ for the executive function score ($N = 147$, three participants were excluded due to an accuracy rate lower than .80). For analysis, incorrect trials (1433 trials, 3.1%) were excluded, as well as latencies below 200ms (32 trials, < 0.01%) and above 1000ms (166 trials, < 0.01%). Additionally, trials 3 SDs above or below each participant's mean were discarded as outliers (533 trials, 1.2%).

## Procedure

Participants were tested between February and December 2019 in a behavioral lab at the Department of Psychology at LMU Munich. After providing written informed consent, participants sat in front of one of 10 computers divided by partitions, in a dimly lit room. All instructions were provided on the computer screen. The software Inquisit 5 Lab [74] was used for task administration. Stimuli were displayed on a 22-inch monitor, viewed from a distance of approximately 65 cm.

Participants first responded to a questionnaire assessing demographic data and exclusion criteria. ANT, VST and SST were administered in random order across participants. Upon completion of all behavior tasks, participants filled in the SPIN. The entire assessment took between 45 and 60 minutes in total.

## Statistical analyses

**Network analysis.** We estimated a partial correlation network with Bonferroni correction (alpha = 0.05). In a network diagram, each node represents a cognitive or symptom variable, whereas each edge represents a significant partial correlation after controlling for the other variables in the variable space. Responses were scaled automatically. We also estimated a network with regularization (graphical lasso), which produced results similar to the partial correlation network. The network analysis was performed using the R packages bootnet [75] and qgraph [76].

## Cluster analysis

All cognitive measures (i.e., ANT, VST, SST; standardized prior to the analysis) were submitted to hierarchical clustering. We used Ward's method on the Euclidian distance, as it has

**Table 1. Descriptives and correlations (N = 147).**

| Variables | M | SD | 1 | 2 | 3 | 4 | 5 | 6 |
|---|---|---|---|---|---|---|---|---|
| 1. ANT: Alt | 49.10 | 22.64 | - | | | | | |
| 2. ANT: Ort | 28.02 | 16.01 | 0.10 | - | | | | |
| 3. ANT: Exc | 68.96 | 20.30 | 0.03 | -0.10 | - | | | |
| 4. VST: Diseng | 22.05 | 91.59 | -0.10 | 0.05 | -0.03 | - | | |
| 5.VST: Engage | -76.74 | 92.97 | 0.06 | 0.03 | -0.05 | -0.17 | - | |
| 6. SST | 0.34 | 0.28 | 0.12 | -0.15 | -0.10 | -0.11 | 0.03 | - |
| 7. SPIN | 23.19 | 8.41 | 0.08 | -0.04 | -0.03 | -0.03 | 0.01 | 0.45* |

ANT: Alt, Ort, Exc = Attentional Network Task: Alerting, Orienting, Executive function; VST: Diseng, Engage = Visual Search Task: Disengagement, Engagement; SST = Scrambled Sentence Task; SPIN = Social Phobia Inventory.

been shown to be superior to other methods of hierarchical clustering [77] and can effectively uncover underlying structures [78]. Analysis was performed using the R package NbClust [79].

## Results

Descriptive information of the sample as well as zero-order correlations are presented in Table 1. The SPIN score had a significant correlation only with the SST score ($r = .45$), but not with the other cognition measures. In line with previous findings [80–83], individuals with higher levels of social anxiety symptoms interpreted ambiguous sentences about social situations more negatively.

### Network analysis

We submitted the observed correlations to network analysis. Both the Bonferroni corrected and the regularized lasso networks (standard tuning parameter of $\gamma = 0.5$) yielded the same result; a network with only one edge. The connection between symptoms of social anxiety (SPIN) and interpretation bias (SST) was positive, i.e., more severe anxiety symptoms were associated with stronger interpretation bias. All remaining associations displayed edges of zero weight, suggesting that attention bias (VST) and attention control (ANT) had no meaningful edge within the network. The regularized network is depicted in Fig 3 (see S3 Fig in S1 File for the Bonferroni correction with identical results).

Within the regularized lasso network, we tested different tuning parameters between 0 and 1; however, results remained unchanged except for a weak negative association between the engagement and disengagement components of attention bias (VST), see S4 Fig in S1 File. Regardless of tuning, we concluded that the only stable connection in a partial correlation network between the observed variables proved to be a positive association between symptoms of social anxiety and interpretation bias. Due to this small number of edges and low edge weights, further analysis of network centrality measures was abandoned as it would not have yielded meaningful results. See S5 Fig in S1 File for edge-weight accuracy.

### Hierarchical clustering

We performed ANOVAs to clarify the group differences for each measure and found statistically significant group differences in each measure except for the VST engagement score. The detailed results can be found in S2 Table (see S1 File). Hierarchical cluster analysis suggested a three-cluster solution (see Fig 4). The number of clusters was determined by visual inspection

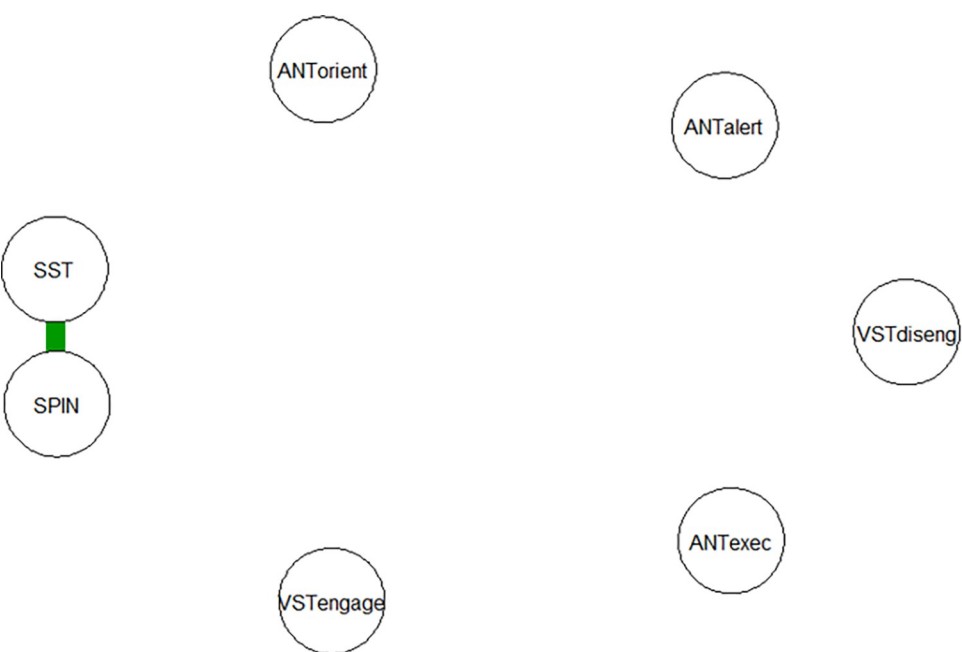

**Fig 3. Diagram of regularized network with tuning parameter set to γ = 0.5.** ANT: Alert, Orient, Execut = Attention Network Test: Alerting, Orienting, Executive function. VST: Diseng, Engage = Visual Search Task: disengagement, engagement; SST = Scrambled Sentence Task; SPIN = Social Phobia Inventory.

of the dendogram (see S6 Fig in S1 File) as well as a set of indices (e.g., Hubert index and D index), the majority of which proposed two or three clusters.

We interpreted these clusters as follows: (1) "Maladaptive Cognition Pattern" ($n$ = 62; 42.2%), (2) "Adaptive Cognition Pattern" ($n$ = 60; 40.8%), and (3) "Distracted Cognition Pattern" ($n$ = 25; 17.0%). The "Maladaptive Cognition Pattern" was characterized by overall poor performances on the ANT (the highest alerting scores indicating high sensitivity for new

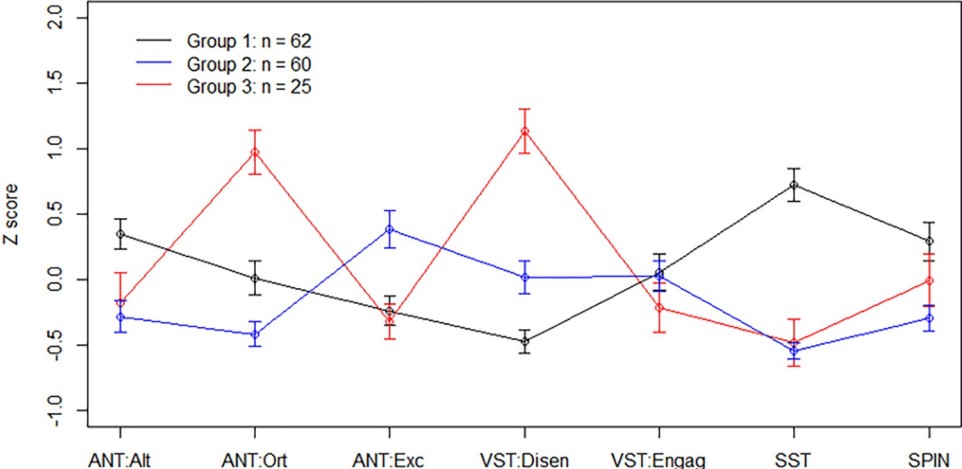

**Fig 4. Profiles of cognitive functions identified by hierarchical clustering.** ANT: Alt, Ort, Exc = Attention Network Test: Alerting, Orienting, Executive function [higher score indicates better functioning in each domain]. VST: Disen, Engag = Visual search task: disengagement, engagement [higher score indicates greater attention bias].
SST = Scrambled Sentence Task [higher score indicates greater interpretation bias]. SPIN = Social Phobia Inventory [higher score indicates greater symptom severity].

stimuli, combined with relatively low scores in executive function) and the highest scores for interpretation bias (SST). While this pattern showed low scores for attention bias (VST), i.e., weaker bias for threatening stimuli, the ANT scores suggest a strong sensitivity and prepared-ness towards new stimuli, which goes along with the highest level of social anxiety among the three patterns. In contrast, the "Adaptive Cognition Pattern" showed similarly low scores for attention bias, but lower alerting scores (i.e., sensitivity) and the highest scores for executive function. In combination with low anxiety symptoms (SPIN), this pattern displayed good cog-nitive functioning without significant biases toward negative stimuli in attention or interpreta-tion, suggesting a potential positive impact of executive function capacities. Lastly, the "Distracted Cognition Pattern" was characterized by the highest scores for disengagement in the VST, i.e., strong difficulty disengaging from angry relative to happy faces, combined with low scores in executive function and interpretation bias. Seeing as this pattern displayed low to mediocre anxiety symptomatology as measured by the SPIN, it may not necessarily relate to anxiety symptoms and rather a general sensitivity toward threatening stimuli.

## Discussion

We performed network and cluster analyses to investigate associations among attention con-trol, cognitive biases, and social anxiety symptoms. Network analysis was applied to find asso-ciations between cognitive (dys)functions and social anxiety symptoms, and hierarchical clustering to identify groups of people with similar cognitive functioning and to relate the identified clusters to social anxiety symptoms.

The network analysis revealed a positive association between social anxiety symptoms (SPIN) and interpretation bias (SST), confirming the notion that more severe anxiety symp-tomatology goes along with more negative interpretations. This is in line with previous studies which found increased negative interpretations in socially anxious compared to non-anxious individuals [see 27 for a meta-analysis]. The positive association between anxiety symptoms and interpretation bias supports the Cognitive Model of Social Phobia by Clark and Wells [1], which postulates a vicious cycle. Biased information processing leads to a significantly more negative perception of social situations, which, in turn, increases fear in those situations and further exacerbates negative interpretations thereof [12]. Thus, ambiguous or mildly negative situations are catastrophized and appear to reinforce socially anxious individuals' negative self-image as well as their belief that others think negatively about them [84, 85].

Contrary to our expectations, however, we found neither a significant association between social anxiety symptoms (SPIN) and attention control (ANT), nor between social anxiety symptoms (SPIN) and attention bias (VST). While an overall bias for emotional stimuli could be revealed, i.e., participants detected emotional targets significantly faster than neutral targets, this bias was not correlated with social anxiety symptoms. These findings are not consistent with previous research presuming a stable correlation between social anxiety and attention bias [21, 22, 57, 86] as well as between social anxiety and altered attention control [42, 43].

However, some previous studies have yielded similarly inconsistent results. Regarding attention bias, socially anxious individuals showed neither significantly faster engagement with nor slower disengagement from socially threatening words [87], angry faces [71, 88], dis-gusted faces [37], or emotional faces in general [89]. Interestingly, the expected attention bias could also not be found in the network study by Heeren and McNally [56], neither for engage-ment with nor for disengagement from fear-relevant stimuli. This may indicate that attention bias is not as relevant to social anxiety as previously assumed. Likewise, with regard to atten-tion control, as noted in the introduction, the ANT has produced an inconsistent pattern of results in relation to social anxiety.

There are several possible explanations for these null associations in the current study. Most critically, both the VST and ANT showed low or modest reliability in our data. This suggests that the scores are highly influenced by measurement error, i.e., the individual differences in attention functioning that should be captured by these tasks are obscured. The rather high error rates and RT exclusions as well as the comparably low reliability could also indicate that some participants were not sufficiently focused during the task. In interpreting the results, this needs to be taken into account. However, it needs to be noted that low reliability is an issue with the majority of measures for cognitive biases [5] and thus a limitation which affects the entire field. Additionally, results could be confounded by the task set-up of the VST. It is conceivable that inter-trial intervals were not sufficiently long in the current study, at 60ms. Participants may not have had enough time to move on from one trial to the next. In attention tasks such as the Stroop Task, shorter inter-trial intervals are associated with slower reactions, higher error rates and diminished self-monitoring [90], as well as increased post-error slowing [91]. Moreover, scoring assumed a bias for negative stimuli over positive stimuli. Instead, there might be a general bias for emotional over neutral stimuli regardless of facial expression [23, 92, 93]. Alternatively, socially anxious individuals may tend to also interpret neutral faces as threatening [94], which might have obscured attention bias. Other possibly confounding variables such as levels of depression [95] were not measured. Regarding attention control (ANT), it could be that anxious individuals do experience a deficit but use other cognitive capacities to compensate, making the deficit harder to detect [96]. Since cognitive performance such as working memory was not measured in this study, this assumption may require further exploration in future studies. Lastly, as pointed out by Leung and colleagues [57], the expected interrelations between different cognitive biases may be easier to detect when biases are measured in one single domain (e.g., facial stimuli only). Since the current study used verbal stimuli to measure interpretation bias and facial stimuli to measure attention bias, transfer between domains may have diminished effects.

In addition to network analysis, exploratory hierarchical clustering suggested that a combination of multiple cognitive measures may be important in investigating cognitive functions in social anxiety. We identified three clusters with "adaptive", "maladaptive", and "distracted" cognitive features. There was a difference in the social anxiety symptoms as measured by the SPIN between the "adaptive" and "maladaptive" cognitive patterns. The fact that the "maladaptive" pattern combines both higher levels of social anxiety symptoms and higher levels of interpretation bias appears consistent with the findings of the network analysis, as it suggests strong interpretation bias as a cognitive phenotype of social anxiety. The differences between the "adaptive" and "maladaptive" cognitive patterns appear to be in line with the assumption that attention control may act as a moderator between cognitive biases and psychopathology [38–40]. With high scores for the executive function component of the ANT, the "adaptive" pattern displays low interpretation bias as well as low anxiety symptoms, suggesting that attention control may be able to positively impact cognitive biases and thus reduce anxiety symptoms. Previous studies have shown that individuals with higher levels of social anxiety are more likely to display attention biases (i.e., initial engagement with threatening stimuli), but attention control may be able to compensate by subsequent disengagement of attention [40]. In this vein, strong capacities in executive function might have a compensatory effect on anxiety [46–48] and possibly limit the impact of threatening stimuli on information processing [40]. This effect in social anxiety has already been replicated [37], yet by measuring attention control via questionnaire rather than experimental task. Accordingly, anxious persons often show deficits in inhibition [5, 97, 98], as can be observed in the ANT scores of the "maladaptive" pattern. This cluster is not only characterized by low executive function, but the highest alerting scores. This is associated with more severe anxiety symptoms, which might indicate that strong alerting

capacities may increase anxiety [46–49]. However, in line with previous studies [42, 47–49], the association between orienting capacities and anxiety symptoms remains inconclusive.

Important to note are limitations of this study. The computational techniques implemented in this study provide an indication of possible associations, but cannot infer direction or temporality of the observed relationships. Network as well as cluster analyses are useful tools in identifying possible links between several variables of interest. However, results remain exploratory and cross-sectional in nature, prohibiting claims on causality. Similarly, one cannot judge the stability of these effects over time, as replicability of networks overall is being debated [99, 100]. Hence, generalizability is limited.

Reliability of all measures is relatively low, with SPIN and VST in particular showing low internal consistency. This limits stability and generalizability of our results, and thus interpretability as well. Low reliability is an inherent problem in cognitive bias research and needs to be addressed in future studies (e.g., by testing reliability more systematically, or developing more reliable tasks such as the Dual Probe Task).

Lastly, results are limited by a relatively homogenous sample. Recruiting participants from the general population rather than exclusively from a clinical sample [56] was meant to increase variance in social anxiety symptoms. However, the sample turned out to consist mostly of highly educated, well-functioning individuals (77% women) with smaller individual differences than we expected initially. Although a third of our sample scored above the cut-off of the SPIN (i.e., experiencing clinically significant levels of anxiety symptoms), more severe levels of symptoms as well as cognitive dysfunctions might not be covered, and thus possible ceiling effects cannot be ruled out. Future studies with analogue samples may consider inducing social-evaluative stress to increase the likelihood of observing effects [5].

## Conclusions

The current study offers a conceptual replication of previous findings regarding the interplay between symptomatology and cognitive functions in social anxiety. These findings confirm the importance of observing several of these variables together rather than separately to better understand the maintenance of symptoms.

A solid association between social anxiety symptoms and interpretation bias confirms the latter as a worthwhile target for therapeutic interventions. The role of attention bias as well as attention control seems less clear. While the current study suggests a compensating effect of executive function capacities, these results are preliminary and future studies should explore this further, ideally via several different paradigms within one design. Results of the cluster analysis show that a wide range of variables can be used to describe cognitive patterns in social anxiety. Since these results are exploratory in nature, however, further studies with a larger sample size are needed to critically examine them and their therapeutic value.

## Supporting information

**S1 File.**
(DOCX)

## Author Contributions

**Conceptualization:** Keisuke Takano, Charlotte E. Wittekind.

**Formal analysis:** Nathalie Claus, Keisuke Takano.

**Investigation:** Nathalie Claus.

**Methodology:** Nathalie Claus, Charlotte E. Wittekind.

**Project administration:** Charlotte E. Wittekind.

**Resources:** Charlotte E. Wittekind.

**Software:** Nathalie Claus.

**Supervision:** Charlotte E. Wittekind.

**Visualization:** Nathalie Claus.

**Writing – original draft:** Nathalie Claus.

**Writing – review & editing:** Keisuke Takano, Charlotte E. Wittekind.

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
