## [Decision Letter · Decision Letter 0]

22 Nov 2022

PONE-D-22-24817The Interplay Between Cognitive Biases, Attention Control, and Social Anxiety Symptoms: A Network and Cluster ApproachPLOS ONE

Dear Dr. Claus,

Thank you for submitting your manuscript to PLOS ONE. After careful consideration, we feel that it has merit but does not fully meet PLOS ONE’s publication criteria as it currently stands. Therefore, we invite you to submit a revised version of the manuscript that addresses the points raised during the review process.

Please note that we have only been able to secure a single reviewer to assess your manuscript. We are issuing a decision on your manuscript at this point to prevent further delays in the evaluation of your manuscript. Please be aware that the editor who handles your revised manuscript might find it necessary to invite additional reviewers to assess this work once the revised manuscript is submitted. However, we will aim to proceed on the basis of this single review if possible.  Please note that the reviewer has concerns about the study analyses and the interpretation of the findings. Please ensure all concerns are carefully addressed.

We look forward to receiving your revised manuscript.

Kind regards,

Alice Coles-Aldridge

Editorial Office

PLOS ONE

Journal Requirements:

Keisuke Takano was supported by Alexander-von-Humboldt foundation.

Reviewers' comments:

Reviewer's Responses to Questions

**Comments to the Author**

1. Is the manuscript technically sound, and do the data support the conclusions?

Reviewer #1: Yes

2. Has the statistical analysis been performed appropriately and rigorously? 

Reviewer #1: Yes

3. Have the authors made all data underlying the findings in their manuscript fully available?

Reviewer #1: Yes

4. Is the manuscript presented in an intelligible fashion and written in standard English?

Reviewer #1: Yes

5. Review Comments to the Author

Reviewer #1: The manuscript "The Interplay Between Cognitive Biases, Attention Control, and Social Anxiety Symptoms: A Network and Cluster Approach" outlines a study designed to investigate the interrelationship between cognitive biases and executive dysfunctions in a network and cluster approach. This study addresses an important topic, in that the causes and maintenance mechanism of social anxiety is increasingly being examined and tested. In general, the study design is clear and is written in good english. However, I do have some concerns regarding the study analyses, and interpretation of findings.

1.The odd-even reliability for the engagement and disengagement score was too low, r = .12 and .08 respectively, which limits the stability and generalizability of the reported results.

2.The results reported in this study are clearly contrary to those of a recent study on a similar topic, and I hope the authors can give a reasonable explanation.

Leung, C. J., Yiend, J., & Lee, T. M. (2022). The Relationship Between Attention, Interpretation, and Memory Bias During Facial Perception in Social Anxiety. Behavior Therapy.

3.The distribution of anxiety scores of the subjects was not provided, such as in a scatter plot. What is the proportion of the number of high and low social anxiety people? Which may affect the explanation of the experimental results.

6. PLOS authors have the option to publish the peer review history of their article (what does this mean?). If published, this will include your full peer review and any attached files.

Reviewer #1: No

---

## [Author Response · Author response to Decision Letter 0]

28 Dec 2022

Copy of enclosed Response Letter:

We would like to thank the editor and the reviewer for the valuable comments. We have considered all suggestions carefully and have done our best to incorporate them. Please find below a detailed description of how we addressed the comments. In addition, we have tracked all changes in the revised manuscript.

REVIEWER: The manuscript "The Interplay Between Cognitive Biases, Attention Control, and Social Anxiety Symptoms: A Network and Cluster Approach" outlines a study designed to investigate the interrelationship between cognitive biases and executive dysfunctions in a network and cluster approach. This study addresses an important topic, in that the causes and maintenance mechanism of social anxiety is increasingly being examined and tested. In general, the study design is clear and is written in good english. However, I do have some concerns regarding the study analyses, and interpretation of findings.

Response: We are very grateful for this positive feedback on our manuscript. 

1. The odd-even reliability for the engagement and disengagement score was too low, r = .12 and .08 respectively, which limits the stability and generalizability of the reported results.

Response: We thank the reviewer for this comment and agree that low reliability is an issue with this current study. However, as we have stated in the manuscript, the field of cognitive bias research generally suffers from reliability issues, pointing towards a need to develop more reliable bias measures. We had been aware of these issues even at the beginning of this project, and thus carefully selected the cognitive-bias measures and tuned the task settings to achieve better reliability (e.g., used the Visual Search Task (VST) instead of a dot-probe task – however, even the VST showed poor reliability in our data). The methodological rigour is, we believe, comparable to the extant studies in the literature, which was, however, not sufficient to find good reliability. Although these results may feel somewhat somber, we argue that it is important to echo the reliability issues, and indeed, the manuscript clearly states the ramifications of low reliability, both in interpreting our results (p. 15) and in discussing limitations (p. 17). To emphasize the reviewer’s point, however, we have adjusted the wording in the limitations section (p. 17):

 “There are several possible explanations for these null associations in the current study. Most critically, both the VST and ANT showed low or modest reliability in our data. This suggests that the scores are highly influenced by measurement error, i.e., the individual differences in attention functioning that should be captured by these tasks are obscured. The rather high error rates and RT exclusions as well as the comparably low reliability could also indicate that some participants were not sufficiently focused during the task. In interpreting the results, this needs to be taken into account. However, it needs to be noted that low reliability is an issue with the majority of measures for cognitive biases (5) and thus a limitation which affects the entire field.”

“Reliability of all measures is relatively low, with SPIN and VST in particular showing low internal consistency. This limits stability and generalizability of our results, and thus interpretability as well. Low reliability is an inherent problem in cognitive bias research and needs to be addressed in future studies (e.g., by testing reliability more systematically, or developing more reliable tasks such as the Dual Probe Task).”

In our opinion, this should not preclude null results from being published. 

2. The results reported in this study are clearly contrary to those of a recent study on a similar topic, and I hope the authors can give a reasonable explanation. Leung, C. J., Yiend, J., & Lee, T. M. (2022). The Relationship Between Attention, Interpretation, and Memory Bias During Facial Perception in Social Anxiety. Behavior Therapy.

Response: We thank the reviewer for pointing out a recent publication relevant to our manuscript. We would rather argue that our results are consistent with this previous study, for the following reasons:

- Leung et al. found that correlations between cognitive bias measures were typically around |r| = 0.10 and no greater than 0.25 (Table 4). Overall, while they did find some statistically (in)direct relationships, sizes of the effects were very small.

- They also documented low reliability for the attention bias measure.

It appears that small (or even null) zero-order correlations between cognitive bias measures are robust and replicable.

We believe, however, that this paper raises additional methodological considerations (e.g., differences in stimuli between measures of different biases, or the possibility of a stress induction when investigating a community sample) which may explain our current results. Accordingly, we have addressed these points in the discussion (p. 16) and limitations (p. 17) sections of our manuscript:

“Lastly, as pointed out by Leung and colleagues (57), the expected interrelations between different cognitive biases may be easier to detect when biases are measured in one single domain (e.g., facial stimuli only). Since the current study used verbal stimuli to measure interpretation bias and facial stimuli to measure attention bias, transfer between domains may have diminished effects.”

“Future studies with analogue samples may consider inducing social-evaluative stress to increase the likelihood of observing effects (5).”

Additionally, we have included the citation in our introduction section.

3. The distribution of anxiety scores of the subjects was not provided, such as in a scatter plot. What is the proportion of the number of high and low social anxiety people? Which may affect the explanation of the experimental results.

Response: We apologize that information on the distribution of social anxiety in our sample was not sufficient. We have added corresponding figures to the supplementary materials and amended the methods section (p. 7) accordingly:

“Cronbach’s alpha for the current sample is α = .72, with scores ranging from 10 to 48 and 33% of the sample scoring above a cut-off of 25 (60), see S1 Fig and S2 Fig in supplementary materials.”

We agree that the right-skewed distribution of our symptom data may have impacted our results and possibly made it more difficult to detect the expected associations. To make this limitation clearer, we have amended the discussion (p. 17) as follows:

“Lastly, results are limited by a relatively homogenous sample. Recruiting participants from the general population rather than exclusively from a clinical sample was meant to increase variance in social anxiety symptoms. However, the sample turned out to consist mostly of highly educated, well-functioning individuals (77% women) with smaller individual differences than we expected initially. Although a third of our sample scored above the cut-off of the SPIN (i.e., experiencing clinically significant levels of anxiety symptoms), more severe levels of symptoms as well as cognitive dysfunctions might not be covered, and thus, possible ceiling effects cannot be ruled out.”

Thank you for taking the time to review our manuscript and the changes we have made based on your suggestions. We believe the reviewer’s input has improved the manuscript and look forward to hearing back from you.

---

## [Decision Letter · Decision Letter 1]

13 Feb 2023

The Interplay Between Cognitive Biases, Attention Control, and Social Anxiety Symptoms: A Network and Cluster Approach

PONE-D-22-24817R1

Dear Dr. Claus,

We’re pleased to inform you that your manuscript has been judged scientifically suitable for publication and will be formally accepted for publication once it meets all outstanding technical requirements.

Kind regards,

Michael B. Steinborn, PhD

Section Editor

PLOS ONE

Additional Editor Comments (optional):

Reviewers' comments:

Reviewer's Responses to Questions

**Comments to the Author**

1. If the authors have adequately addressed your comments raised in a previous round of review and you feel that this manuscript is now acceptable for publication, you may indicate that here to bypass the “Comments to the Author” section, enter your conflict of interest statement in the “Confidential to Editor” section, and submit your "Accept" recommendation.

Reviewer #1: All comments have been addressed

2. Is the manuscript technically sound, and do the data support the conclusions?

Reviewer #1: Yes

3. Has the statistical analysis been performed appropriately and rigorously? 

Reviewer #1: Yes

4. Have the authors made all data underlying the findings in their manuscript fully available?

Reviewer #1: Yes

5. Is the manuscript presented in an intelligible fashion and written in standard English?

Reviewer #1: Yes

6. Review Comments to the Author

Reviewer #1: I have no more comments.

7. PLOS authors have the option to publish the peer review history of their article (what does this mean?). If published, this will include your full peer review and any attached files.

Reviewer #1: No

---

## [Editor Report · Acceptance letter]

22 Feb 2023

PONE-D-22-24817R1 

The Interplay Between Cognitive Biases, Attention Control, and Social Anxiety Symptoms: A Network and Cluster Approach 

Dear Dr. Claus:

I'm pleased to inform you that your manuscript has been deemed suitable for publication in PLOS ONE. Congratulations! Your manuscript is now with our production department. 

Kind regards, 

on behalf of

Dr. Michael B. Steinborn 

Section Editor

PLOS ONE